# Cross-lingual Name Tagging and Linking for 282 Languages

## Abstract

The ambitious goal of this work is to develop a cross-lingual name tagging and linking framework for 282 languages that exist in Wikipedia. Given a document in any of these languages, our framework is able to identify name mentions, assign a coarse-grained or fine-grained type to each mention, and link it to an English Knowledge Base (KB) if it is linkable. We achieve this goal by performing a series of new KB mining methods: generating "silver-standard" annotations by transferring annotations from English to other languages through cross-lingual links and KB properties, refining annotations through self-training and topic selection, deriving language-specific morphology features from anchor links, and mining word translation pairs from cross-lingual links. Both name tagging and linking results for 282 languages are promising on Wikipedia data. The results on non-Wikipedia data (news articles and discussion forum posts) are also comparable to the supervised models trained from manually annotated documents including thousands of names. All the data sets, resources and systems for 282 languages will be made publicly available as a new benchmark.

## 1 Introduction

Information provided in languages which people can understand saves lives in crises. For example, language barrier was one of the main difficulties faced by humanitarian workers responding to the Ebola crisis in 2014. We propose to break language barriers by extracting information (e.g., entities) from a massive variety of languages and ground the information into an existing knowledge base which is accessible to a user in his/her own language (e.g., a reporter from the World Health Organization who speaks English only).

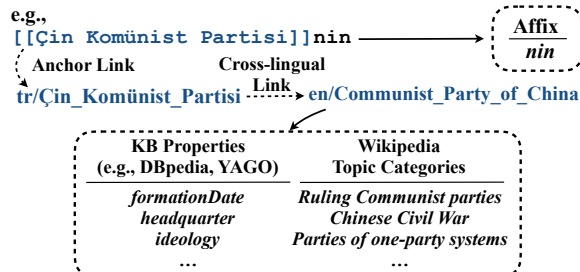

Figure 1: Examples of Wikipedia Markups and KB Properties

It is probably too ambitious and challenging to perform this task for all the languages in the world due to the scarcity of resources, but perhaps we could give it a try for hundreds of them. For instance, Wikipedia is a massively multi-lingual resource that currently hosts 295 languages and contains naturally annotated markups [1] and rich informational structures through crowd-sourcing for 35 million articles in 3 billion words. Name mentions in Wikipedia are often labeled as anchor links to their corresponding referent pages. Each entry in Wikipedia is also mapped to external knowl-

---

[1] https://en.wikipedia.org/wiki/Help:Wiki_markup

edge bases such as DBpedia[2], YAGO (Mahdisoltani et al., 2015) and Freebase (Bollacker et al., 2008) that contain rich properties. Figure 1 shows an example of Wikipedia markups and KB properties. We will leverage these markups for developing a cross-lingual name tagging and linking system for 282 languages. The major challenges and our proposed new solutions are summarized as follows.

**Creating "Silver-standard" through cross-lingual entity transfer.** The first step is to classify English Wikipedia entries into certain entity types and then propagate these labels to other languages. Most of the previous work (Nothman et al., 2008; Dakka and Cucerzan, 2008; Ringland et al., 2009; Alotaibi and Lee, 2012; Nothman et al., 2012; Althobaiti et al., 2014) manually classified many English Wikipedia entries into pre-defined ACE [3] or CoNLL entity types[4]. In contrast, we exploit the English Abstract Meaning Representation (AMR) corpus (Banarescu et al., 2013) which includes both name tagging and linking annotations for fine-grained entity types to train an automatic classifier. We also map each AMR type to YAGO type hierarchy. Therefore, the generated typing schema has multiple levels of granularity. Furthermore, we exploit each entry's properties in DBpedia as features and thus eliminate the need of language-specific features and resources such as part-of-speech tagging as in previous work. Then we project annotations to all entity mentions in Wikipedia articles in other languages through cross-lingual links (Section 2.2).

**Refine annotations through self-training.** However, such initial annotations are too incomplete and inconsistent. Previous work used name string match to propagate labels. In contrast, we apply self-training to label other mentions without links in Wikipedia articles even if they have different surface forms from the linked mentions (Section 2.3).

**Customize annotations through cross-lingual topic transfer.** For some populous languages, we obtain millions of labeled sentences from self-training. For the first time, we propose to customize the annotations for specific down-stream applications. Again, we use a cross-lingual knowledge transfer strategy to leverage the widely available English corpora to choose entities with specific Wikipedia topic categories (Section 2.4).

**Derive morphology analysis from Wikipedia markups.** Another unique challenge for many morphologically rich languages is to segment each token into its stemming form and affixes. Previous methods relied on either high-cost supervised learning (Roth et al., 2008; Mahmoudi et al., 2013; Ahlberg et al., 2015), or low-quality unsupervised learning (Gronroos et al., 2014; Ruokolainen et al., 2016). We exploit Wikipedia markups to automatically learn affixes as language-specific features (Section 2.5).

**Mine word translations from cross-lingual links.** Name translation is a crucial step to generate candidate entities in cross-lingual entity linking. Only a small percentage of names can be directly translated by matching against cross-lingual Wikipedia title pairs. Based on the observation that Wikipedia titles within any language tend to follow a consistent style and format, we propose an effective method to derive word translation pairs from these titles based on automatic alignment (Section 3.2).

## 2 Name Tagging

### 2.1 Overview

Our first step is to generate "silver-standard" name annotations from Wikipedia markups and train a universal name tagger. Figure 2 shows our overall procedure and the following subsections will elaborate each component.

### 2.2 Annotation Generation

We start by assigning an entity type or "other" to each English Wikipedia entry. We utilize Abstract Meaning Representation (AMR) corpus (Banarescu et al., 2013) where each entity name mention is manually labeled as one of 139 types and linked to Wikipedia if it's linkable. In total we obtain 2,756 entity mentions, along with their AMR entity types, Wikipedia titles and DBPedia properties.

For each pair of AMR entity type $t^a$ and YAGO entity type $t^y$, we compute the Pointwise Mutual Information (Church and Hanks, 1990) of mapping $t^a$ to $t^y$ across all mentions in the AMR corpus. In this way, our framework produces three levels of entity typing schemas with different granularity: 4 regular types (Person (PER), Organization (ORG), Geo-political Entity (GPE), Location

---

[2]http://wiki.dbpedia.org
[3]http://www.itl.nist.gov/iad/mig/tests/ace/
[4]http://www.cnts.ua.ac.be/conll2003/ner/

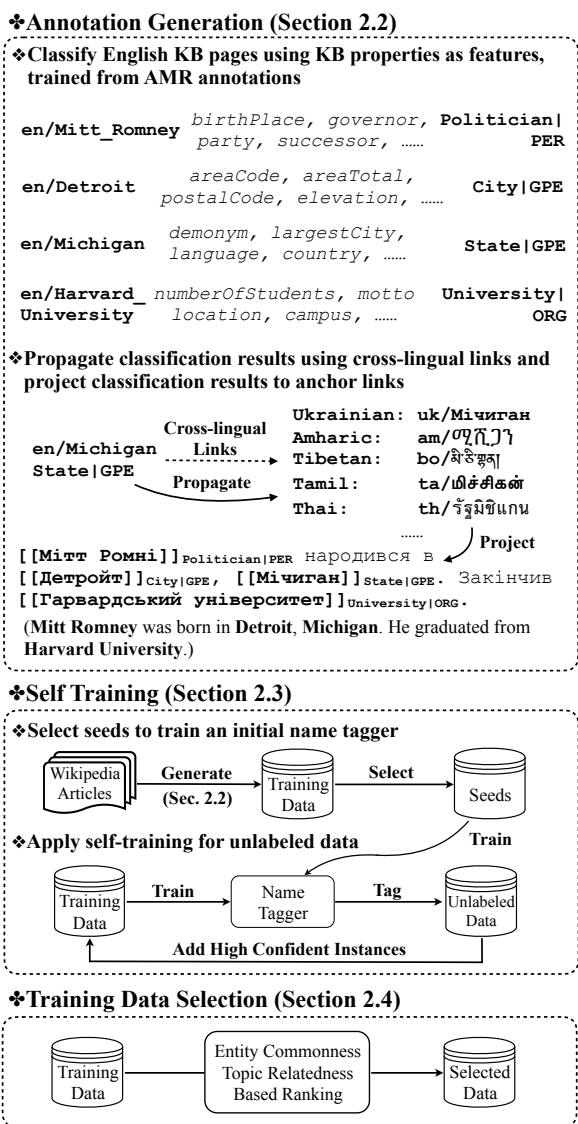

Figure 2: Name Tagging Annotation Generation and Training

(LOC)), 139 types in AMR, and 9,154 types in YAGO.

We also propose a new idea to leverage an entity's properties in DBpedia as features for assigning types. For example, an entity with a birth date is likely to be a person, while an entity with a population property is likely to be a geo-political entity. Using entity properties as features, we train Maximum Entropy models to assign types with three levels of granularity to all English Wikipedia pages. In total we obtained 10 million English pages labeled as entities of interest.

Nothman et al. (2012) manually annotated 4,853 English Wikipedia pages with 6 coarse-grained types (Person, Organization, Location, Other, Non-Entity, Disambiguation Page). Using this data set for training and testing, we achieved 96.0% F-score on this initial step, slightly better than their results (94.6% F-score).

Next, we propagate the label of each English Wikipedia page to all entity mentions in all languages in the entire Wikipedia through monolingual redirect links and cross-lingual links.

## 2.3 Self-Training

The name annotations acquired from the above procedure are far from complete to compete with manually labeled gold-standard data. For example, if a name mention appears multiple times in a Wikipedia article, only the first mention is labeled with an anchor link. We apply self-training to propagate and refine the labels.

We use a typical neural network architecture that consists of Bi-directional Long Short-Term Memory and Conditional Random Fields (CRFs) network (Lample et al., 2016) as our underlying learning model for the name tagger. We first train an initial name tagger using seeds selected from the labeled data. We adopt an idea from (Guo et al., 2014) which computes normalized pointwise mutual information (NPMI) (Bouma, 2009) between a tag and a token:

$$NPMI(tag, token) = \frac{\ln \frac{p(tag,token)}{p(tag)p(token)}}{-\ln p(tag, token)} \quad (1)$$

Then we select the sentences in which all annotations satisfy $NPMI(tag, token) > \tau$ as seeds [5].

For all Wikipedia articles in a language, we cluster the unlabeled sentences into $n$ clusters [6] by collecting sentences with low cross-entropy into the same cluster. Then we apply the initial tagger to the first unlabeled cluster, select the automatically labeled sentences with high confidence, add them back into the training data, and then re-train the tagger. This procedure is repeated $n$ times until we scan through all unlabeled data.

## 2.4 Training Data Selection

For some populous languages that have many millions of pages in Wikipedia, we obtain many sentences from self-training. In some emergent settings such as natural disasters it's important to train a system rapidly. Therefore we develop the following effective methods to rank and select high-quality annotated sentences.

---

[5] $\tau = 0$ in our experiment.

[6] $n = 20$ in our experiment.

**Commonness**: we prefer sentences that include common entities appearing frequently in Wikipedia. We rank names by their frequency and dynamically set the frequency threshold to select a list of common names. We first initialize the name frequency threshold $S$ to 40. If the number of the sentences is more than a desired size $D$ for training [7], we set the threshold $S = S + 5$, otherwise $S = S - 5$. We iteratively run the selection algorithm until the size of the training set reaches $D$ for a certain $S$.

**Topical Relatedness**: Various criteria should be adopted for different scenarios. Using an emergent disaster setting as a use case, we prefer sentences that include entities related to disaster related topics. We run an English name tagger (Manning et al., 2014) and entity linker (Pan et al., 2015) on the Leidos corpus released by the DARPA LORELEI program [8]. The Leidos corpus consists of documents related to various disaster topics. Based on the linked Wikipedia pages, we rank the frequency of Wikipedia categories and select the top 1% categories (4,035 in total) for our experiments. Some top-ranked topic labels include "*International medical and health organizations*", "*Human rights organizations*", "*International development agencies*", "*Western Asian countries*", "*Southeast African countries*"and "*People in public health*". Then we select the annotated sentences including names (e.g., "*World Health Organization*") in all languages labeled with these topic labels.

### 2.5 KB Derived Features

When a Wikipedia user tries to link an entity mention in a sentence to an existing page, she/he will mark the title (the entity's canonical form, without affixes) within the mention using brackets "`[[]]`", from which we can naturally derive a word's stem and affixes for free. For example, from the Wikipedia markups of the following Turkish sentence: "`Kıta Fransası, güneyde `**`[[Akdeniz]]den`**` kuzeyde [[Manş Denizi]] ve `**`[[Kuzey Denizi]]ne,`** `doğuda `**`[[Ren Nehri]]nden`**` batıda `**`[[Atlas Okyanusu]]na`**` kadar yayılan topraklarda yer alır.` *(Metropolitan France extends from the Mediterranean Sea to the English Channel and the North Sea, and from the Rhine to the Atlantic*

Ocean.*)*", we can learn the following suffixes: "*den*", "*ne*", "*nden*" and "*na*". We use such affix lists to perform basic word stemming, and use them as additional features to determine name boundary and type. For example, "*den*" is a noun suffix which indicates ablative case in Turkish. `[[Akdeniz]]den` means "*from Mediterranean Sea*". Thus, if a token contains a suffix "*den*", it is likely to be a location name. Note that this approach can only perform morphology analysis for words whose stem forms and affixes are directly concatenated.

Table 1 summarizes name tagging features.

| Features | Descriptions |
|---|---|
| Form | Lowercase forms of $(w_{-1}, w_0, w_{+1})$ |
| Case | Case of $w_0$ |
| Syllable | The first and the last character of $w_0$ |
| Stem | Stems of $(w_{-1}, w_0, w_{+1})$ |
| Affix | Affixes of $(w_{-1}, w_0, w_{+1})$ |
| Gazetteer | Cross-lingual gazetteers learned from training data |
| Embeddings | Character embeddings and word embeddings [9]learned from training data |

Table 1: Name Tagging Features

## 3 Cross-lingual Entity Linking

### 3.1 Overview

After we extract names from documents in a source language, we try to translate them into English by automatically mined word translation pairs (Section 3.2), and then link translated English mentions to an external English KB (in this paper we use DBpedia) (Section 3.3). The overall linking process is illustrated in Figure 3.

### 3.2 Name Translation

If the source language is not English, an important step in the linking process is to translate identified name mentions into English. We can utilize cross-lingual Wikipedia title pairs for exact name string matching. For example, "*Pekin*" and "*Beijing*" is a Turkish-English Wikipedia title pair. However, our experiments as well as previous work (Nothman et al., 2012) show that only a small percentage of name mentions can be covered in this way. Fortunately, these cross-lingual title pairs, generated through crowd-sourcing, generally follow a consistent style and format in each language. For example, from Table 2 we can see that the order of

---

[7] $D$ = 30,000 in our experiment.

[8] http://www.darpa.mil/program/low-resource-languages-for-emergent-incidents

[9] For languages that don't have space, we consider each character as a token, and use character embeddings only.

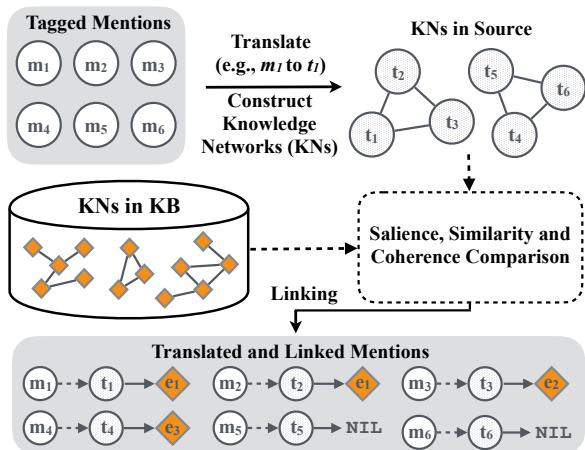

Figure 3: Cross-lingual Entity Linking Overview

| Extracted Cross-lingual Wikipedia Title Pairs | |
|---|---|
| *"Pekin"* | |
| **Pekin** | Beijing |
| **Pekin** metrosu | Beijing Subway |
| **Pekin** Ulusal Stadyumu | Beijing National Stadium |
| *"Teknoloji"* | |
| Nükleer **teknoloji** | Nuclear technology |
| **Teknoloji** transferi | Technology transfer |
| **Teknoloji** eğitimi | Technology education |
| *"Enstitüsü"* | |
| Torchwood **Enstitüsü** | Torchwood Institute |
| Hudson **Enstitüsü** | Hudson Institute |
| Smolny **Enstitüsü** | Smolny Institute |
| *"Pekin Teknoloji"* [NONE] | |
| *"Teknoloji Enstitüsü"* | |
| Kraliyet **Teknoloji Enstitüsü** | Royal Institute of Technology |
| Karlsruhe **Teknoloji Enstitüsü** | Karlsruhe Institute of Technology |
| Georgia **Teknoloji Enstitüsü** | Georgia Institute of Technology |
| *"Pekin Teknoloji Enstitüsü"* [NONE] | |

| Mined Word Translation Pairs | | |
|---|---|---|
| Word | Translation | Alignment Confidence |
| *pekin* | **Beijing** | *Exact Match* |
| | beijing | 0.5263 |
| | peking | 0.3158 |
| *teknoloji* | **technology** | 0.8890 |
| | technology | 0.0233 |
| | technological | 0.0174 |
| *enstitüsü* | **institute** | 0.2765 |
| | of | 0.2028 |
| | for | 0.0221 |

Table 2: Word Translation Mining from Cross-lingual Wikipedia Title Pairs

modifier and head word keeps consistent in Turkish and English titles. Therefore, we propose a new and simple method to mine word and phrase translation pairs as follows.

For each name mention, we generate all possible combinations of continuous tokens. For example, no Wikipedia titles contain the Turkish name string "*Pekin Teknoloji Enstitüsü (Beijing Institute of Technology)*". We generate the following 6 combinations: "*Pekin*", "*Teknoloji*", "*Enstitüsü*", "*Pekin Teknoloji*", "*Teknoloji Enstitüsü*" and "*Pekin Teknoloji Enstitüsü*". We then extract all cross-lingual Wikipedia title pairs containing each combination. Finally we run GIZA++ (Och and Ney, 2003) to extract word for word translations from these title pairs. Using this exhaustive search method we can obtain many word translations as shown in Table 2.

### 3.3 Entity Linking

Given a set of tagged name mentions $M = \{m_1, m_2, ..., m_n\}$, we first obtain their English translations $T = \{t_1, t_2, ..., t_n\}$ using the approach described above. Then we apply an unsupervised collective inference approach to link $T$ to the KB, similar to some previous work (Kulkarni et al., 2009; Fernandez et al., 2010; Radford et al., 2010; Cucerzan, 2011; Han and Sun, 2011; Ratinov et al., 2011; Chen and Ji, 2011; Kozareva et al., 2011; Dalton and Dietz, 2013; Pan et al., 2015; Wang et al., 2015). We construct knowledge networks (KNs) $g(t_i)$ for $T$ based on their co-occurrence within a context window [10]. For each translated name mention $t_i$, an initial list of candidate entities $E(t_i) = \{e_1, e_2, ..., e_k\}$ is generated based on a surface form dictionary mined from KB properties (e.g., *redirects*, *names*, *aliases*). If no surface form can be matched then we determine the mention as unlinkable. Then we construct knowledge network $g(e_j)$ for each entity candidate $e_j$ in $t_i$'s entity candidate list $E(t_i)$. We compute the similarity between $g(t_i)$ and $g(e_j)$ based on three measures as described in (Pan et al., 2015): salience, similarity and coherence, and select the candidate entity with the highest score as the appropriate entity for linking.

## 4 Experiments

### 4.1 Performance on Wikipedia Data

We first conduct an evaluation using Wikipedia data as "silver-standard". For each language, we use 70% of the selected sentences for training and 30% for testing. For entity linking, we don't have ground truth for unlinkable mentions, so we only

---

[10]In our experiments, we use the previous four and next four name mentions as a context window.

compute linking accuracy for linkable name mentions. Table 3 presents the overall performance for three coarse-grained entity types: person, organization and geo-political entity/location.

Not surprisingly, name tagging performs better for languages with more mentions. The f-score is generally higher than 80% when there are more than 30K mentions, and it significantly drops when there are less than 250 mentions. The languages with low name tagging performance can be categorized into three types: (1) the number of mentions is less than 2K, such as Atlantic-Congo (Wolof), Berber (Kabyle), Chadic (Hausa), Oceanic (Fijian), Hellenic (Greek), Igboid (Igbo), Mande (Bambara), Kartvelian (Georgian, Mingrelian), Timor-Babar (Tetum), Tupian (Guarani) and Iroquoian language groups; Precision is generally higher than recall for most of these languages, because the small number of linked mentions is not enough to cover a wide variety of entities. (2) there is no space between words, including Chinese, Thai and Japanese; (3) they are not written in latin script, such as the Dravidian group (Tamil, Telugu, Kannada, Malayalam).

The training instances for various entity types are quite imbalanced for many languages. For example, Cebuano data includes 1,251 person names, 787,349 geo-political entity/location names and 8,292 organization names. As a result, the performance of organization is the lowest, while geo-political entities and locations achieve higher than 75% F-scores for most languages. The linking accuracy is higher than 80% for most languages.

Also note that since we don't have perfect annotations on Wikipedia data for any language, these results can be used to estimate how predictable our "silver-standard" data is, but they are not directly comparable to traditional name tagging results measured against gold-standard data annotated by human.

## 4.2 Performance on Non-Wikipedia Data

Therefore, in order to have more direct comparison with state-of-the-art name taggers trained from human annotated gold-standard data, we conduct experiments on Non-Wikipedia data in 9 languages for which we have human annotated ground truths from the DARPA LORELEI pro-

---

[11]The mapping to language names can be found at https://meta.wikimedia.org/wiki/List_of_Wikipedias

gram. Table 4 shows the data statistics. The documents are from news sources and discussion fora.

We can see that our approach advances state-of-the-art language-independent methods (Zhang et al., 2016a; Tsai et al., 2016) on the same data sets for most languages, and achieves 5.8% - 17.6% lower F-scores than the models trained from manually annotated gold-standard documents that include thousands of name mentions. We also measured the impact of our morphology analysis methods based on the affix lists derived from Wikipedia markups on two morphologically-rich languages: Turkish and Uzbek. The morphology features contributed 11.1% and 7.1% absolute F-score gains to Turkish and Uzbek respectively.

### 4.3 Impact of Self-Training

Using Turkish as a case study, the learning curves of self-training on Wikipedia and Non-Wikipedia test sets are shown in Figure 4. We can see that self-training provides significant improvement for both Wikipedia (6% absolute gain) and Non-Wikipedia test data (12% absolute gain). As expected the learning curve on Wikipedia data is more smooth and converges more slowly than that of Non-Wikipedia data.

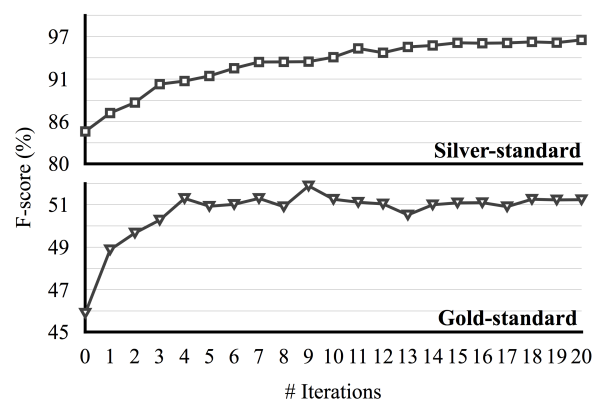

Figure 4: Learning Curve of Self-training

## 5 Related Work

Our work was mainly inspired from previous work that leveraged Wikipedia markups to train name taggers (Nothman et al., 2008; Dakka and Cucerzan, 2008; Mika et al., 2008; Ringland et al., 2009; Alotaibi and Lee, 2012; Nothman et al., 2012; Althobaiti et al., 2014). Our approach doesn't need any manual annotations or language-specific features, while generates both coarse-grained and fine-grained types.

| L | M | F | A | L | M | F | A | L | M | F | A | L | M | F | A |
|---|---|---|---|---|---|---|---|---|---|---|---|---|---|---|---|
| en | 70M | 91.8 | 85.3 | lb | 102K | 81.5 | 88.4 | nah | 8.4K | 89.9 | 90.6 | m-b | 1.6K | 78.3 | 81.3 |
| de | 14M | 89.0 | 88.4 | tl | 102K | 92.7 | 90.3 | b-s | 8.1K | 84.5 | 88.0 | rw | 1.6K | 95.4 | 92.7 |
| fr | 12M | 93.3 | 92.4 | sq | 97K | 94.1 | 92.9 | rm | 8.0K | 82.0 | 91.3 | ln | 1.6K | 82.8 | 93.3 |
| es | 6.9M | 93.9 | 93.4 | sco | 94K | 86.8 | 89.6 | azb | 7.9K | 88.4 | 90.6 | kl | 1.5K | 75.0 | 95.5 |
| it | 6.9M | 96.6 | 94.7 | ce | 90K | 99.4 | 71.0 | ay | 7.9K | 88.5 | 96.2 | sn | 1.5K | 95.0 | 99.3 |
| ja | 6.5M | 79.2 | 91.8 | pnb | 81K | 90.8 | 86.2 | ps | 7.7K | 66.9 | 91.8 | av | 1.4K | 82.0 | 83.7 |
| ru | 6.4M | 90.1 | 89.2 | als | 81K | 85.0 | 91.2 | mi | 7.5K | 95.9 | 24.5 | wo | 1.3K | 87.7 | 97.3 |
| pl | 6.0M | 90.0 | 91.3 | lmo | 80K | 98.3 | 89.0 | gag | 7.3K | 89.3 | 84.0 | xal | 1.3K | 98.7 | 90.9 |
| nl | 5.6M | 93.2 | 91.9 | is | 76K | 80.2 | 83.2 | am | 7.0K | 84.7 | 83.0 | na | 1.2K | 87.6 | 88.7 |
| sv | 4.8M | 93.6 | 89.7 | fy | 75K | 86.6 | 91.4 | nrm | 7.0K | 96.4 | 99.6 | ltg | 1.2K | 74.3 | 92.1 |
| pt | 3.9M | 90.7 | 92.4 | an | 73K | 93.0 | 93.1 | nso | 6.9K | 98.9 | 99.4 | tet | 1.2K | 73.5 | 92.3 |
| sh | 2.9M | 97.8 | 95.6 | gu | 70K | 76.0 | 92.4 | sc | 6.8K | 78.1 | 95.2 | ksh | 1.2K | 56.0 | 83.6 |
| uk | 2.3M | 91.5 | 89.4 | qu | 63K | 92.5 | 88.2 | so | 6.5K | 85.8 | 95.7 | haw | 1.1K | 88.0 | 84.6 |
| zh | 2.1M | 82.0 | 89.7 | ast | 60K | 89.2 | 88.4 | co | 6.0K | 85.4 | 89.9 | pfl | 1.1K | 42.9 | 80.4 |
| fi | 2.0M | 93.4 | 94.7 | z-y | 58K | 87.3 | 88.4 | stq | 6.0K | 70.0 | 92.6 | tpi | 1.1K | 83.3 | 90.1 |
| cs | 1.9M | 94.6 | 92.0 | sw | 56K | 93.4 | 93.0 | pcd | 5.8K | 86.1 | 75.8 | lo | 1.0K | 52.8 | 92.0 |
| no | 1.9M | 94.1 | 93.8 | ga | 55K | 85.3 | 91.3 | dsb | 5.8K | 84.7 | 82.1 | ki | 1.0K | 97.5 | 90.0 |
| hu | 1.9M | 95.9 | 92.8 | jv | 54K | 82.6 | 87.8 | wa | 5.8K | 81.6 | 82.0 | ty | 1.0K | 86.7 | 89.8 |
| ca | 1.8M | 90.3 | 91.6 | new | 51K | 98.2 | 97.9 | wuu | 5.8K | 79.7 | 90.8 | ady | 979 | 92.7 | 95.5 |
| fa | 1.4M | 96.4 | 83.8 | nds | 44K | 84.5 | 88.2 | frr | 5.7K | 70.1 | 86.3 | ig | 968 | 74.4 | 45.5 |
| sr | 1.4M | 95.3 | 93.3 | ht | 42K | 98.9 | 99.8 | lad | 5.6K | 92.3 | 95.6 | tyv | 903 | 91.1 | 98.8 |
| ko | 1.3M | 90.6 | 91.4 | pms | 40K | 98.0 | 89.5 | z-c | 5.4K | 88.2 | 87.0 | tn | 902 | 76.9 | 90.1 |
| tr | 1.2M | 90.5 | 87.3 | bar | 39K | 97.1 | 96.3 | nap | 4.9K | 86.9 | 88.6 | cu | 898 | 75.5 | 76.9 |
| vi | 1.2M | 89.6 | 82.0 | kn | 39K | 60.1 | 92.1 | crh | 4.9K | 90.1 | 89.9 | sm | 888 | 80.0 | 85.3 |
| ro | 1.1M | 90.6 | 85.0 | ba | 38K | 93.8 | 86.3 | km | 4.6K | 52.2 | 89.9 | to | 866 | 92.3 | 69.2 |
| he | 1.0M | 79.0 | 92.5 | arz | 35K | 77.8 | 89.3 | c-z | 4.5K | 75.0 | 89.2 | tum | 831 | 93.8 | 98.5 |
| ar | 1.0M | 88.3 | 85.3 | ne | 31K | 81.5 | 92.5 | lez | 4.4K | 84.2 | 82.3 | r-r | 750 | 93.0 | 90.5 |
| bg | 871K | 65.8 | 88.4 | cv | 31K | 95.7 | 97.4 | mai | 4.3K | 99.7 | 90.0 | om | 709 | 74.2 | 81.1 |
| ceb | 796K | 96.3 | 86.6 | fo | 30K | 83.6 | 78.1 | hak | 4.3K | 85.5 | 88.1 | glk | 688 | 59.5 | 80.8 |
| sk | 779K | 87.3 | 90.3 | li | 28K | 89.4 | 91.6 | ang | 4.2K | 84.0 | 96.7 | lbe | 651 | 88.9 | 68.6 |
| id | 775K | 87.8 | 87.3 | ckb | 28K | 88.1 | 76.1 | r-t | 4.2K | 88.1 | 89.0 | bjn | 640 | 64.7 | 89.5 |
| da | 717K | 87.1 | 85.8 | gd | 28K | 92.8 | 91.7 | udm | 4.2K | 88.9 | 85.0 | srn | 619 | 76.5 | 90.1 |
| z-m | 680K | 99.3 | 89.2 | io | 26K | 87.2 | 95.5 | csb | 4.1K | 87.0 | 94.4 | mdf | 617 | 82.2 | 97.3 |
| eo | 647K | 88.7 | 81.4 | mn | 25K | 76.4 | 84.4 | lij | 4.1K | 72.3 | 92.0 | tw | 572 | 94.6 | 90.4 |
| eu | 640K | 82.5 | 88.3 | tg | 25K | 88.3 | 90.6 | nov | 4.0K | 77.0 | 95.0 | pih | 555 | 87.2 | 90.0 |
| ms | 569K | 86.8 | 84.1 | bug | 25K | 99.9 | 90.0 | ace | 4.0K | 81.6 | 90.9 | rmy | 551 | 68.5 | 90.0 |
| sl | 537K | 89.5 | 90.1 | scn | 25K | 93.2 | 89.2 | gn | 4.0K | 71.2 | 89.3 | lg | 530 | 98.8 | 89.9 |
| el | 516K | 84.6 | 88.3 | ku | 23K | 83.2 | 85.1 | koi | 4.0K | 89.6 | 93.5 | chr | 530 | 70.6 | 90.9 |
| hr | 490K | 82.8 | 88.5 | pa | 21K | 74.8 | 84.3 | mhr | 3.9K | 86.7 | 94.7 | ha | 517 | 75.0 | 80.0 |
| et | 456K | 86.8 | 89.9 | yi | 20K | 76.9 | 87.2 | min | 3.8K | 85.8 | 88.5 | got | 506 | 91.7 | 92.3 |
| lt | 423K | 86.3 | 87.2 | sa | 20K | 73.9 | 93.0 | ext | 3.7K | 77.8 | 91.6 | ab | 506 | 60.0 | 93.3 |
| nn | 393K | 88.1 | 86.7 | hsb | 20K | 91.5 | 74.9 | kab | 3.3K | 75.7 | 84.3 | bi | 490 | 88.5 | 93.3 |
| th | 391K | 56.2 | 87.7 | vls | 19K | 78.2 | 89.1 | szl | 3.0K | 82.7 | 92.7 | st | 455 | 84.4 | 89.8 |
| gl | 382K | 87.4 | 88.2 | ilo | 18K | 90.3 | 93.1 | tk | 2.9K | 86.3 | 90.1 | chy | 450 | 85.1 | 89.9 |
| war | 369K | 94.9 | 87.4 | vec | 15K | 87.9 | 93.8 | kv | 2.9K | 89.7 | 97.2 | iu | 450 | 66.7 | 83.3 |
| sim | 368K | 85.7 | 91.6 | bpy | 15K | 98.3 | 98.0 | f-v | 2.9K | 65.4 | 88.8 | zu | 449 | 82.3 | 89.9 |
| lv | 349K | 92.1 | 89.8 | my | 13K | 51.5 | 92.4 | gan | 2.9K | 84.9 | 90.9 | pnt | 445 | 61.5 | 87.5 |
| ur | 309K | 96.4 | 83.4 | mzn | 13K | 86.4 | 86.9 | fur | 2.8K | 84.5 | 89.2 | ik | 436 | 94.1 | 54.3 |
| ka | 290K | 79.8 | 89.5 | os | 13K | 87.4 | 89.4 | mwl | 2.7K | 76.1 | 89.4 | lrc | 416 | 65.2 | 90.0 |
| hy | 280K | 90.4 | 88.9 | ky | 13K | 71.8 | 88.4 | nv | 2.7K | 90.9 | 94.1 | bm | 386 | 77.3 | 87.5 |
| vo | 269K | 98.5 | 97.5 | xmf | 13K | 73.4 | 94.4 | sd | 2.7K | 65.8 | 90.9 | za | 382 | 57.1 | 90.3 |
| uz | 266K | 98.3 | 94.8 | or | 13K | 86.4 | 73.5 | bxr | 2.6K | 75.0 | 91.9 | mo | 373 | 69.6 | 90.2 |
| la | 260K | 90.8 | 89.4 | pam | 13K | 87.2 | 92.9 | bo | 2.6K | 70.4 | 88.9 | ss | 362 | 69.2 | 71.4 |
| mk | 236K | 93.4 | 83.3 | si | 12K | 88.7 | 94.1 | frp | 2.5K | 86.2 | 98.5 | ee | 297 | 63.2 | 90.0 |
| bs | 227K | 84.8 | 89.9 | sah | 12K | 91.2 | 78.8 | myv | 2.5K | 88.6 | 94.4 | dz | 262 | 50.0 | 80.0 |
| kk | 224K | 88.3 | 81.8 | mt | 12K | 82.3 | 90.3 | cdo | 2.5K | 91.0 | 96.2 | ak | 258 | 86.8 | 92.6 |
| be | 205K | 84.1 | 88.3 | mrj | 12K | 97.0 | 55.3 | gom | 2.4K | 88.8 | 90.0 | sg | 245 | 100.0 | 89.9 |
| ta | 193K | 77.9 | 88.2 | n-n | 11K | 92.6 | 92.9 | bh | 2.2K | 92.6 | 92.8 | ts | 236 | 93.3 | 88.9 |
| az | 176K | 85.1 | 86.0 | as | 11K | 89.6 | 83.0 | ug | 2.1K | 79.7 | 92.4 | rn | 185 | 40.0 | 75.0 |
| hi | 170K | 86.9 | 88.0 | vep | 11K | 85.8 | 89.8 | kaa | 2.1K | 55.2 | 89.5 | ve | 183 | 100.0 | 83.3 |
| cy | 169K | 90.7 | 89.3 | diq | 10K | 79.3 | 80.9 | krc | 2.1K | 84.9 | 88.9 | ny | 169 | 56.0 | 75.0 |
| bn | 168K | 93.8 | 87.2 | zea | 10K | 86.8 | 90.3 | dv | 2.0K | 76.2 | 94.3 | ff | 168 | 76.9 | 88.9 |
| af | 142K | 85.7 | 91.1 | hif | 10K | 81.1 | 96.7 | rue | 1.9K | 82.7 | 72.1 | ch | 159 | 70.6 | 90.0 |
| br | 138K | 87.0 | 85.5 | se | 9.7K | 90.3 | 83.5 | eml | 1.8K | 83.5 | 88.5 | xh | 141 | 35.3 | 75.0 |
| mg | 134K | 98.7 | 92.9 | ie | 9.5K | 88.8 | 95.5 | arc | 1.8K | 68.5 | 82.0 | fj | 126 | 75.0 | 80.0 |
| te | 128K | 80.5 | 86.1 | su | 9.4K | 72.7 | 89.2 | pdc | 1.8K | 78.1 | 91.1 | ks | 124 | 75.0 | 83.3 |
| b-x | 126K | 85.1 | 87.7 | bcl | 9.3K | 82.3 | 94.0 | kbd | 1.7K | 74.9 | 80.6 | ti | 52 | 94.2 | 90.0 |
| ml | 125K | 82.4 | 88.8 | yo | 9.2K | 94.0 | 93.0 | pap | 1.7K | 88.8 | 59.1 | cr | 49 | 91.8 | 89.8 |
| tt | 119K | 87.7 | 91.4 | ia | 8.9K | 75.4 | 90.5 | jbo | 1.7K | 92.4 | 96.5 | pi | 41 | 83.3 | 90.2 |
| oc | 118K | 92.5 | 90.0 | kw | 8.7K | 94.0 | 96.1 | pag | 1.7K | 91.2 | 89.5 | | | | |
| mr | 109K | 82.4 | 91.0 | gv | 8.4K | 84.8 | 89.1 | kg | 1.6K | 82.1 | 90.1 | | | | |

Table 3: Performance on Wikipedia Data (L: language ID [11]; M: the number of name mentions in selected annotated data; F: name tagging F-score (%); A: linking accuracy (%))

| Language | *Gold* Training | *Silver* Training | Test |
|---|---|---|---|
| Bengali | 8,760 | 22,093 | 3,495 |
| Hungarian | 3,414 | 34,022 | 1,320 |
| Russian | 2,751 | 35,764 | 1,213 |
| Tamil | 7,033 | 25,521 | 4,632 |
| Tagalog | 4,648 | 15,839 | 3,351 |
| Turkish | 2,172 | 37,058 | 1,353 |
| Uzbek | 3,137 | 64,242 | 2,056 |
| Vietnamese | 2,261 | 63,971 | 987 |
| Yoruba | 4,061 | 9,274 | 3,395 |

Table 4: # of Names in Non-Wikipedia Data

| Language | Training from *Gold* | Training from *Silver* | (Zhang et al., 2016a) | (Tsai et al., 2016) |
|---|---|---|---|---|
| Bengali | 61.6 | **44.0** | 34.8 | 43.3 |
| Hungarian | 63.9 | 47.9 | - | - |
| Russian | 61.8 | 49.4 | - | - |
| Tamil | 42.2 | **35.7** | 33.8 | 29.6 |
| Tagalog | 70.7 | 58.3 | 51.3 | 65.4 |
| Turkish | 57.3 | **51.5** | 43.6 | 47.1 |
| Uzbek | 56.0 | 44.2 | - | - |
| Vietnamese | 54.3 | 44.5 | - | - |
| Yoruba | 55.1 | **37.6** | 36.0 | 36.7 |

Table 5: Name Tagging F-score (%) on Non-Wikipedia Test Data

Many fine-grained entity typing approaches (Fleischman and Hovy, 2002; Giuliano, 2009; Ekbal et al., 2010; Xiao and Weld, 2012; Yosef et al., 2012; Nakashole et al., 2013; Gillick et al., 2014; Yogatama et al., 2015; Corro et al., 2015) also created annotations based on Wikipeida anchor links. Our framework performs both name identification and typing and takes advantage of richer structures in the KBs. Previous work on Arabic name tagging (Althobaiti et al., 2014) extracted entity titles as a gazetteer for stemming, and thus it cannot handle unknown names. We developed a new method to derive generalizable affixes for morphologically rich language based on Wikipedia markups.

Wikipedia has also been used as additional features to improve various Information Extraction (IE) tasks, including name tagging (Kazama and Torisawa, 2007), coreference resolution (Ponzetto and Strube, 2006), relation extraction (Chan and Roth, 2010) and event extraction (Hogue and Nothman, 2014). Other automatic name annotation generation methods have been proposed, including KB driven distant supervision (An et al., 2003; Mintz et al., 2009; Ren et al., 2015) and cross-lingual projection (Li et al., 2012; Kim et al., 2012; Che et al., 2013; Wang et al., 2013; Wang

and Manning, 2014; Zhang et al., 2016b).

Some recent research (Zhang et al., 2016a; Littell et al., 2016; Tsai et al., 2016) under the DARPA LORELEI program focused on developing name tagging techniques for low-resource languages. These approaches require English annotations for projection (Tsai et al., 2016), some input from a native speaker, either through manual annotations (Littell et al., 2016), or a linguistic survey (Zhang et al., 2016a). Without using any manual annotations, our name taggers outperform previous methods on the same data sets for many languages. NIST TAC-KBP Tri-lingual entity linking (Ji et al., 2016) focused on three languages: English, Chinese and Spanish. (McNamee et al., 2011) extended it to 21 languages. But their methods required labeled data and name transliteration. We share the same goal as (Sil and Florian, 2016) to extend cross-lingual entity linking to all languages in Wikipedia. They exploited Wikipedia links to train a supervised linker. We mine reliable word translations from cross-lingual Wikipedia titles, which enables us to adopt unsupervised English entity linking techniques.

## 6 Conclusions and Future Work

We developed a simple yet effective framework that can extract names from 282 languages and link them to an English KB. This framework follows a fully automatic training and testing pipeline, without the needs of any manual annotations or knowledge from native speakers. We evaluated our framework on both Wikipedia articles and external formal and informal texts and obtained promising results. To the best of our knowledge, our multilingual name tagging and linking framework is applied to the largest number of languages. We release the following resources for each of these 282 languages: "silver-standard" name tagging and linking annotations with multiple levels of granularity, morphology analyzer if it's a morphologically-rich language, and an end-to-end name tagging and linking system. In this work, we treat all languages independently when training their corresponding name taggers. In the future, we will explore the topological structure of related languages and exploit cross-lingual knowledge transfer to enhance the quality of extraction and linking. The general idea of deriving noisy annotations from KB properties can also be extended to other IE tasks such as relation extraction.

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
