# Peer review of "Cross-lingual Name Tagging and Linking for 282 Languages"

_ACL 2017 — decision unknown_

[Official Review · Reviewer 1 · rating 4 · confidence 4]
soundness 5 · originality 3 · clarity 4 · impact 3 · substance 5 · appropriateness 5 · meaningful comparison 4 · presentation format Oral Presentation

- Strengths:
   - The paper states clearly the contributions from the beginning 
   - Authors provide system and dataset
   - Figures help in illustrating the approach
   - Detailed description of the approach
   - The authors test their approach performance on other datasets and compare
to other published work

- Weaknesses:
   -The explanation of methods in some paragraphs is too detailed and there is
no mention of other work and it is repeated in the corresponding method
sections, the authors committed to address this issue in the final version.
   -README file for the dataset [Authors committed to add README file]

- General Discussion:
   - Section 2.2 mentions examples of DBpedia properties that were used as
features. Do the authors mean that all the properties have been used or there
is a subset? If the latter please list them. In the authors' response, the
authors explain in more details this point and I strongly believe that it is
crucial to list all the features in details in the final version for clarity
and replicability of the paper. 
   - In section 2.3 the authors use Lample et al. Bi-LSTM-CRF model, it might
be beneficial to add that the input is word embeddings (similarly to Lample et
al.)
   - Figure 3, KNs in source language or in English? (since the mentions have
been translated to English). In the authors' response, the authors stated that
they will correct the figure.
   - Based on section 2.4 it seems that topical relatedness implies that some
features are domain dependent. It would be helpful to see how much domain
dependent features affect the performance. In the final version, the authors
will add the performance results for the above mentioned features, as mentioned
in their response. 
   - In related work, the authors make a strong connection to Sil and Florian
work where they emphasize the supervised vs. unsupervised difference. The
proposed approach is still supervised in the sense of training, however the
generation of training data doesn’t involve human interference

[Official Review · Reviewer 2 · rating 4 · confidence 4]
soundness 5 · originality 3 · clarity 3 · impact 3 · substance 5 · appropriateness 5 · meaningful comparison 4 · presentation format Poster

- Strengths:

- Very impressive resource

- fully automatic system - particularly suitable for cross-lingual learning
across many languages

- Good evaluation both within and outside wikipedia. Good comparison to works
that employed manual resources.

- Weaknesses:

- The clarity of the paper can be improved.

- General Discussion:

This paper presents "a simple yet effective framework that can extract names
from 282 languages and link them to an English KB". Importantly, the system is
fully automatic, which is particularly important when aiming to learn across
such a large number of languages. Although this is far from trivial, the
authors are able to put their results in context and provide evaluation both
within and outside of wikipedia - I particularly like the way the put their
work in the context of previous work that uses manual resources, it is a good
scientific practice and I am glad they do not refrain from doing that in worry
that this would not look good.

The clarity of the paper can improve. This is not an easy paper to write due to
the quite complex process and the very large scale resource it generates.
However, the paper is not very well organized and at many points I felt that I
am reading a long list of details. I encourage the authors to try and give the
paper a better structure. As one example, I would be happy to see a better
problem definition and high level motivations from the very beginning. Other
examples has to do with better exposition of the motivations, decisions and
contributions in each part of the paper (I admire the efforts the authors have
already made, but I think this can done even better). This is an important
paper and it deserves a clearer presentation.

All in all I like the paper and think it provides an important resource. I
would like to see this paper presented in ACL 2017.